biochemistry/biotechnology

microalgae, inorganic carbon, nitrogen and phosphorus recycle, lipid accumulation, bioenergy

**Author for correspondence:**
Yun Duan
e-mail: duanyun@tyut.edu.cn

This article has been edited by the Royal Society of Chemistry, including the commissioning, peer review process and editorial aspects up to the point of acceptance.

# Nutrients recycle and the growth of *Scenedesmus obliquus* in synthetic wastewater under different sodium carbonate concentrations

Yun Duan, Xin Guo, Jingjing Yang, Mingmei Zhang and Yangyang Li

College of Environmental Science and Engineering, Taiyuan University of Technology, Taiyuan, Shanxi 030024, People's Republic of China

YD, 0000-0002-1914-4181; XG, 0000-0002-7196-6393

This study illustrated the growth of *Scenedesmus obliquus* and recycle of nutrients in wastewater combined with inorganic carbon under autotrophic conditions. *Scenedesmus obliquus* was cultivated under different conditions by adding sodium carbonate ($Na_2CO_3$) at 15–40 mg l$^{-1}$ separately in wastewater containing high nitrogen and phosphorus content. The growth characteristics of *S. obliquus*, pH and dissolved inorganic carbon (DIC) changes of microalgae liquid, the recycle rate of ammonia and phosphorus and lipid content were determined. The changes of pH and DIC showed that *S. obliquus* could use $Na_2CO_3$ to grow, with lipid contents of 18–25%. Among all $Na_2CO_3$ concentrations, 20 mg l$^{-1}$ was the optimum, of which *S. obliquus* had the highest $NH_3$-N recycle of 52% and $PO_4^{3-}$-P recycle of 67%. By the 14th day, its biomass production also reaches the maximum of 0.21 g l$^{-1}$. However, inorganic carbon fixation rate was inversely proportional to its concentration. Moreover, the biomass was in positive correlation with the $Na_2CO_3$ concentration except 20 mg l$^{-1}$, which provided a possibility that *S. obliquus* could be acclimatized to adjust to high concentrations of inorganic carbon to promote biomass accumulation and recycle of nutrients.

## 1. Introduction

The synergetic coupling of water treatment and producing bioenergy have attracted increasing attention, contributing to solve

eutrophication, improve water quality and alleviate the cost of cultivating microalgae. If disposed without sufficient treatment, nitrogen and phosphorus in all kinds of wastewater would adversely affect environmental quality as well as people's lives [1]. Nitrogen and phosphorus can be recycled as important nutrients for microalgae growth. Microalgae produce a variety of biological substances, such as peptides, proteins, enzymes, chlorophylls, energy transfer molecules, genetic materials and lipids [1], and provide lots of energy. $CO_2$ in air and even flue gas can be used as inorganic carbon source for algae cultivation to ease the greenhouse effect. Microalgae-based wastewater treatment has obvious advantages such as rapid growth rate, high photosynthetic efficiency and energy-saving [2–4]. Furthermore, microalgae do not require high-quality agricultural land compared with crops, thus avoiding possible competition with food or production [3].

Algae strain is also a major factor to be considered during microalgae-based treatment. In order to remove nitrogen and phosphorus efficiently and increase algal growth, it is critical to select the appropriate algae strains. *Scenedesmus obliquus* is a freshwater microalga with great potential for commercial applications and extraordinary vitality in various wastewaters since it grows fast, tolerates a wide range of temperature and pH and is easy to cultivate [5,6]. Thus far, there have been several attempts aimed to treat various types of wastewater based on *S. obliquus*, such as piggery slurry and urban wastewater [6,7]. Wang *et al.* [7] found that six microalgae including *S. obliquus* (FACHB-12) treated with the method based on UV irradiation followed by gradual domestication not only grew well in undiluted slurry, but achieved the removal rate of about 90% total nitrogen and 86% total phosphorus. Martínez *et al.* [6]. cultivated *S. obliquus* in urban sewage waters where this organism has shown great vitality and efficiency in eliminating N and P compounds.

Many factors affecting the algal properties, such as cultivation modes, carbon sources, N/P ratio, heavy metals, light wavelengths, light intensities and light–dark cycles [3,8–13] should be considered. There are three major cultivation modes, namely photoautotrophic, heterotrophic and autotrophic cultivation. By the synergistic effect of photoautotrophic and heterotrophic metabolism, autotrophic growth may be widely used in the near future. It was found that Cu(II) could inhibit $NH_3$-N and total phosphorus (TP) removal by *Coelastrella* sp. cultured in anaerobically digested swine wastewater (ADSW), and when Cu(II) was higher than 1.0 mg l$^{-1}$, microalgae biomass stopped increasing [9]. Zhou *et al.* [10] further investigated that the mechanisms of the interactions on nutrients removal and role of dissolved organic matter released from the duckweed systems on microcosmic migration of heavy metals, and reached the optimal $Cu^{2+}$ concentration of 0.96 mg l$^{-1}$ for the removal of $NH_3$-N and TP in swine wastewater. Most studies focused on the effects of organic carbon sources, such as glucose and xylose [8,14], and inorganic carbon such as carbon dioxide [11] on growth of microalgae in an autotrophic mode. Considering the solubility, the temperature of the flue gas, the tolerance and the transportation costs of $CO_2$, $CO_2$ in the flue gas can be transferred into the alkaline solution in form of ions for microalgae cultivation in practical application. The optimum content of carbon dioxide can be determined by determining the amount of inorganic carbon in the form of ions. However, very few studies have considered the effect of $Na_2CO_3$ whose composition are $HCO_3^-$ and $CO_3^{2-}$, which is equivalent to aerating carbon dioxide into microalgae cultivation system.

In this paper, under autotrophic condition, nitrogen and phosphorus with high concentrations in wastewater were recycled by *S. obliquus* using different concentrations of $Na_2CO_3$ as inorganic carbon (IC) source. In order to determine the role of $Na_2CO_3$, pH and dissolved inorganic carbon (DIC) of microalgae liquid were measured during the growth cycle. Meanwhile, the growth of microalgae and the removal of nutrients were studied, which might provide a theoretical basis for the practical application of *S. obliquus*.

# 2. Material and methods

## 2.1. Microalgae strain and culture conditions

The microalgae *S. obliquus* were obtained from Shanghai Guangyu Biological Technology. It was preserved in BG11 medium [13]: $NaNO_3$ (1500 mg l$^{-1}$), $K_2HPO_4 \cdot 3H_2O$ (40 mg l$^{-1}$), $MgSO_4 \cdot 7H_2O$ (75 mg l$^{-1}$), $CaCl_2 \cdot 2H_2O$ (36 mg l$^{-1}$), $Na_2CO_3$ (20 mg l$^{-1}$) and ferric ammonium citrate (6 mg l$^{-1}$), citric acid (6 mg l$^{-1}$), $Na_2EDTA \cdot 2H_2O$ (1 mg l$^{-1}$), and 1 ml l$^{-1}$ of trace elements solution consisting of $H_3BO_3$ (2.86 mg l$^{-1}$), $MnCl_2 \cdot 4H_2O$ (1.81 mg l$^{-1}$), $ZnSO_4 \cdot 7H_2O$ (0.22 mg l$^{-1}$), $Na_2MoO_4 \cdot 2H_2O$ (0.39 mg l$^{-1}$), $CuSO_4 \cdot 5H_2O$ (0.08 mg l$^{-1}$) and $Co(NO_3)_2 \cdot 6H_2O$ (0.05 mg l$^{-1}$).

A seed culture was prepared in a 250 ml baffled culture flask containing 200 ml of the BG11 medium and then inoculated with 20 ml microalgae solution in a ratio of 1 : 10, initial biomass concentration controlled at $33 \times 10^5$ cells ml$^{-1}$. The microalgae in exponential growth phase were used in the study. All the operations were carried out in sterile conditions. The cultivation was in 25°C incubator,

illuminated by 15 W LEDs tubes at average light intensity 3000 lux measured by spectroradiometer and changed position regularly to ensure illumination uniform. The applied photoperiod was 12 h : 12 h of white : dark light controlled by an automatic timer switch. Experiment was conducted with duplicates shaken twice per day and set up three parallel experiments for each concentration.

## 2.2. Preparation of synthetic wastewater

Original effluent made it difficult for microalgae to absorb nutrients and to grow; furthermore, high turbidity considerably reduced light intensity in the medium, which also inhibited the growth of photosynthetic algae [15]. So in this paper, synthetic wastewater with high nitrogen and phosphorus concentration was used to simulate the environment of high ammonia nitrogen wastewater. The synthetic wastewater was based on BG11 medium. And in order to investigate the effects of ammonium and orthophosphate, 0.4 g l$^{-1}$ of NH$_4$Cl and 0.077 g l$^{-1}$ of K$_2$HPO$_4$ were used instead of the nitrogen source and phosphorus source in BG11 culture medium, respectively, with other compositions the same. Ammonium was used as nitrogen source because the uptake of ammonium is important in microalgae nitrogen recycle and nitrogen often exists as ammonium in wastewater, especially for livestock wastewater and anaerobically digested wastewater [16].

Batch experiments were performed in 250 ml flasks with 200 ml synthetic wastewater. Biomass concentration was controlled at around OD$_{680}$ (optical density at 680 nm of microalgae liquid) of 0.2 after inoculation. For the experiments on the effects of Na$_2$CO$_3$ concentration, except the control group (0 mg l$^{-1}$), the final concentrations of six samples were 15, 20, 25.0, 30.0, 35.0 and 40.0 mg l$^{-1}$. The culture mediums were sterilized in an autoclave at 121°C for 20 min before inoculation. After inoculation, the initial nitrogen and phosphorus concentrations were measured to be 151 mg l$^{-1}$, 10.46 mg l$^{-1}$, respectively. All other conditions remained the same.

## 2.3. Water quality monitoring

A volume of 10 ml microalgae suspension was collected every 2 days from each medium for microalgae growth and nutrient recycle analysis. The samples were first centrifuged at 8000 r.p.m. for 5 min, after which the supernatants were filtered using a 0.45 μm cellulose membrane filter. Then, the filtrates were appropriately diluted and analysed for pH (PHS-3C type pH meter), carbonate alkalinity (Calk) (Acid-base titration), ammonia nitrogen (NH$_3$-N) (Nash-reagent spectrophotometric method) and phosphorus (PO$_4^{3-}$-P) (molybdenum–antimony anti-spectrophotometric method). All data were measured in parallel three times and averaged. The percentage recycle was obtained using the following equation:

$$\text{Percentage removal} = 100\% \times \frac{C_s - C_f}{C_s}, \tag{2.1}$$

where $C_s$ and $C_f$ are defined as the mean values of nutrient concentration at initial start time and finish time, respectively.

According to the following method [17], calculating DIC concentration from Calk and pH at 25°C:

$$[\text{DIC}] = \frac{\text{Calk} + [\text{H}^+] - [\text{OH}^-]}{(\alpha_1 + 2\alpha_2)}, \tag{2.2}$$

where parameters $\alpha_1$ and $\alpha_2$ are the dissociation fractions of HCO$_3^-$ and H$_2$CO$_3$ respectively, as calculated by equations (2.3) and (2.4),

$$\alpha_1 = \frac{1}{1 + [\text{H}^+]/K_1 + K_2/[\text{H}^+]}, \tag{2.3}$$

$$\alpha_2 = \frac{1}{1 + [\text{H}^+]/K_2 + [\text{H}^+]/K_1/K_2}, \tag{2.4}$$

where $K_1$ and $K_2$ are the first and the second composite acidity constants of H$_2$CO$_3$ respectively.

## 2.4. Dry weight and lipid determination

After *S. obliquus* grew for 14 days, the microalgae cells were harvested through centrifugation at the end of the experiment. The total lipid content was extracted with chloroform/methanol (1/1, v/v) and

quantified gravimetrically. The growth of microalgae was monitored by measuring optical density at 680 nm ($OD_{680}$) with UV–Vis spectrophotometer (UV-1900PC) (Shanghai, China). The specific growth rate $\mu$ of microalgae growth was measured by using the following equation:

$$\mu(\text{day}^{-1}) = \frac{\ln(N_1/N_2)}{t_2 - t_1},$$ (2.5)

where $N_1$ and $N_2$ are defined as dry biomass (g l$^{-1}$) at time $t_1$ and $t_2$, respectively. The weight showed a linear relationship with $OD_{680}$, the detail is as follows:

$$\text{dry weight (g l}^{-1}) = 0.294 \times OD_{680}, R^2 = 0.998.$$ (2.6)

# 3. Results and discussion

## 3.1. Autotrophic growth of Scenedesmus obliquus at different Na$_2$CO$_3$ concentrations

The growth characteristics of S. obliquus at different $Na_2CO_3$ concentrations within 14 days under autotrophic conditions were investigated as shown in figure 1. The results clearly showed that, by the 14th day of cultivation, the maximum biomass production was 0.21 g l$^{-1}$ when the $Na_2CO_3$ concentration was 20 mg l$^{-1}$, far bigger than that at other $Na_2CO_3$ concentrations, and the specific growth rate was also the same. The reason might be that microalgae could quickly adapt to the environment of 20 mg l$^{-1}$ $Na_2CO_3$ concentration, which was same as BG11 medium. The growth curves of other concentrations were comparable in the first 4 days, but when the $Na_2CO_3$ concentration was 15 mg l$^{-1}$, the biomass concentration almost no longer rose in later period, probably because of the insufficient carbon source in culture conditions. The growth curves of $Na_2CO_3$ concentrations of 25, 30, 35 and 40 mg l$^{-1}$ were similar, indicating that the growth of S. obliquus was promoted. In addition, it was found that more biomass was produced with the increase of $Na_2CO_3$ concentration. Perhaps microalgae could be acclimatized to adjust to conditions contained high concentration IC.

The growth of S. obliquus was subjected to various restrictions, including nutrients, light intensity and space. Like other microorganisms, microalgae growth can undergo four growth phases: lag, exponential, stationary and lysis [18]. There was no lag phase in the study. Maximum specific growth rate ($\mu_{max}$) was always achieved at the beginning of cultivation [5] on day 2 in this study. The parameter was closely related to the growth of microalgae, and curves at all concentrations showed similar trends, namely, the greater the growth rate, the faster the microalgae grew. As the microalgae growth time went on, the specific growth rate declined. The highest $\mu_{max}$ value (0.266 d$^{-1}$) was obtained with the addition of 20 mg l$^{-1}$ $Na_2CO_3$ concentration, the specific growth rates were reduced to 0.01 d$^{-1}$ in the end. The results illustrated that microalgae reached a stable period or even into the decay period after 6 days due to various restrictions of conditions, therefore, microalgae grew slowly at the end of the experiment.

## 3.2. Absorption and utilization of Na$_2$CO$_3$ by photosynthesis of microalgae

According to the literature [19], the relationship between the composition and pH in the carbonic acid solution is shown in figure 2. It can be seen that when the pH was 8.3, the components in the carbonate solution existed in the form of $HCO_3^-$. Due to $HCO_3^-$ being absorbed by microalgae, the pH gradually increased to about 11 later in the experiment. By that time, $CO_3^{2-}$ accounted for 75% of the dissolved IC in the solution, which was difficult to be used by microalgae cells.

The results in figure 3 describe that pH of the samples changed with culture time under the different $Na_2CO_3$ concentrations. As can be seen from figure 3, pH showed the trend of rising firstly and then falling. In detail, the initial pH was positively correlated with the $Na_2CO_3$ concentration, and furthermore, the pH increased significantly in the first 4 days after inoculation, due to the microalgae species being in an exponential growth period. Then the rising trend gradually levelled off, meaning the microalgae species subsequently entered a stationary period. Meanwhile, the pH was pushed to around 11. After 12 days, the pH began to decline, indicating that the microalgae entered a lysis period. The rise of pH was mainly due to the carbon sequestration of photosynthesis, demonstrating that the IC in the solution was consumed. Therefore, S. obliquus could grow well when the concentration of $Na_2CO_3$ was 15–40 mg l$^{-1}$.

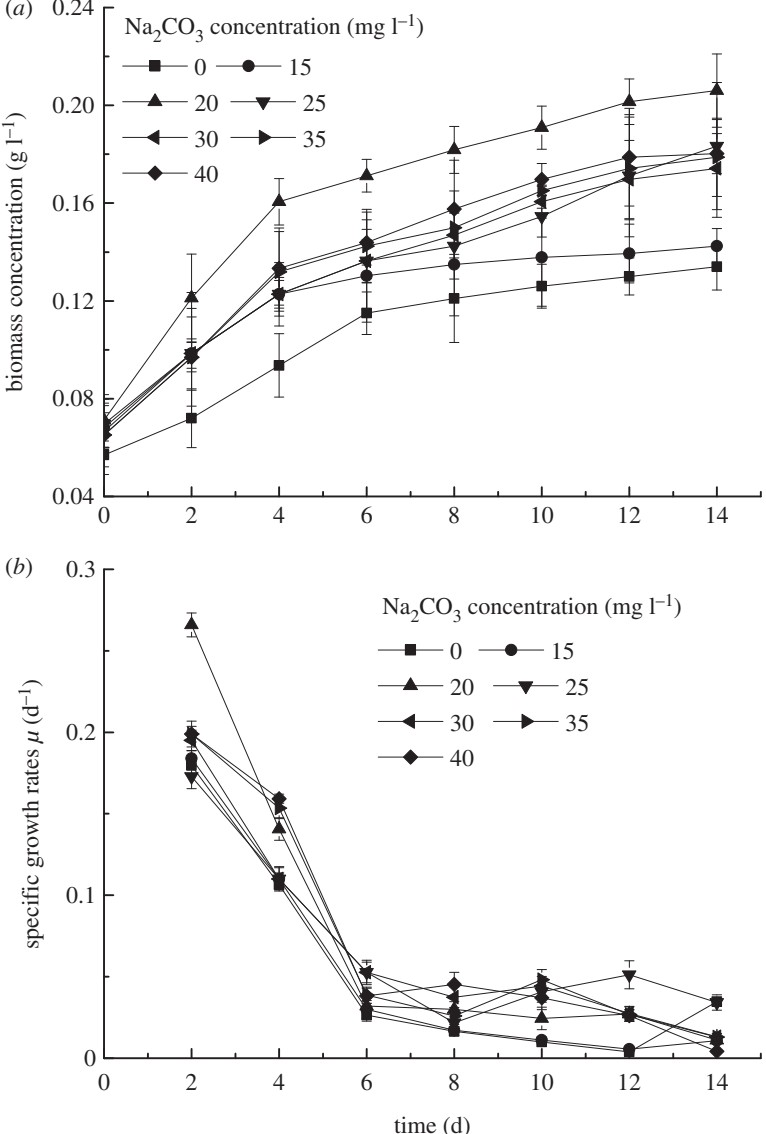

**Figure 1.** The growth curve (*a*) and special growth rate (*b*) of *S. obliquus* at six $Na_2CO_3$ concentrations under autotrophic condition.

In weakly alkaline media, the IC forms that microalgae can use are mainly $HCO_3^-$ and $CO_2$ dissolved in water. In particular, the increase of pH is owing to the absorption of $HCO_3^-$ by microalgae, leading to $OH^-$ accumulation. The mechanism is as follows:

$$HCO_3^- + H_2O \overset{CA}{=} H_2CO_3 + OH^- \tag{3.1}$$

and

$$H_2CO_3 \overset{CA}{=} CO_2 + H_2O, \tag{3.2}$$

where CA represents the extracellular carbonic anhydrase.

No matter what form of carbonate, only converted into $CO_2$ by CA could it be used by microalgae cells. Under extracellular CA catalysis, $HCO_3^-$ dehydrated to $CO_2$, then transferring from the outside of the cell to the intracellular environment, accelerating $CO_2$ formation and supply. These captured carbons are transported through the chloroplast membrane proteins into the chloroplasts for photosynthetic cooperation to promote biomass growth and intracellular constituent synthesis. Besides the consumption of IC, the absorption of other nutrients can also lead to the alkalization of the solution environment. The majority essential nutrients existed in ionic form, such as $HCO_3^-$, $NO_3^-$ and $H_2PO_4^-$, which

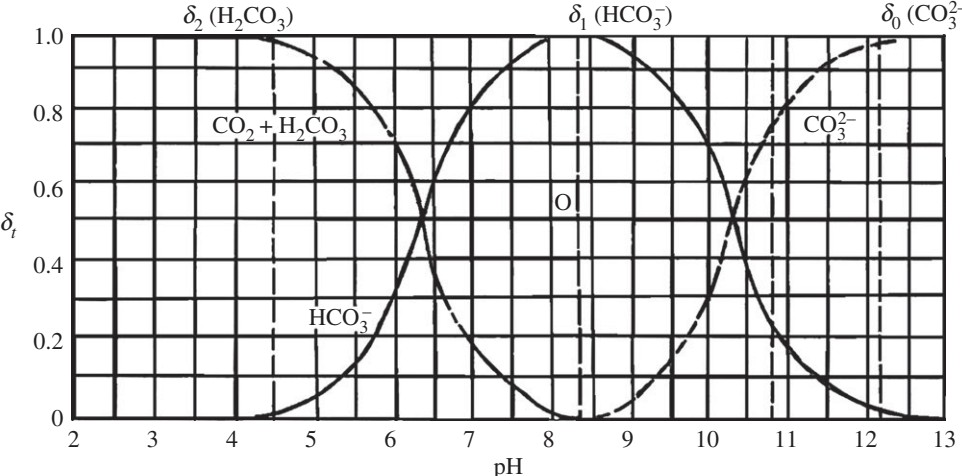

**Figure 2.** δ-pH graph in carbonated solution.

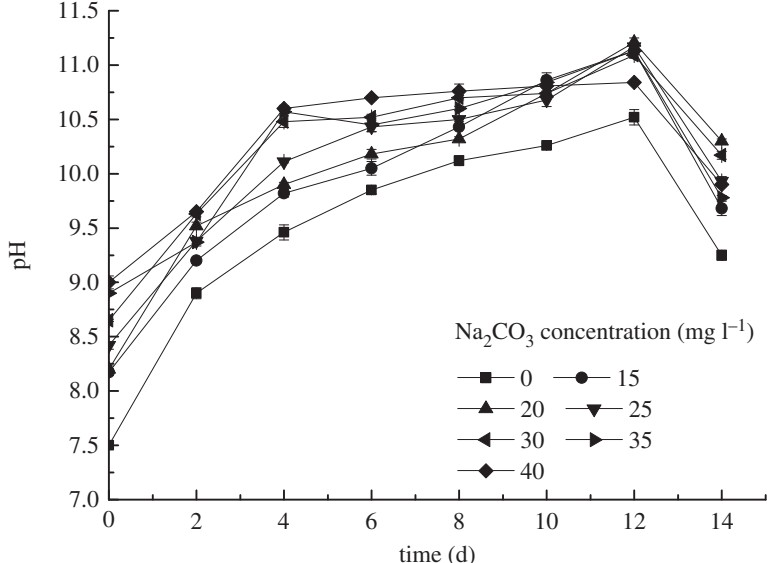

**Figure 3.** The pH changes of the microalgae liquid under the different $Na_2CO_3$ concentrations.

assimilated $H^+$ from culture medium and released $OH^-$, thereby leading to pH increase [20]. The optimal C/N ratio for microalgae growth is 100 : 18 [21], while in this study, the synthetic wastewater was with high nitrogen and phosphorus. And the concentrations of nitrogen and phosphorus were much higher than that of carbonate, which was sufficient for the microalgae growth. It can be seen from the above that the concentration of nitrogen and phosphorus decreased sharply in the early stage and stabilized until the sixth day. Correspondingly, it can be observed that from day 4, pH was no longer significantly different than before. With the consumption of nutrients, cells growth entered a stable phase. It is worth noting that the pH of sample with initial 20 mg $l^{-1}$ $Na_2CO_3$ still maintained a relatively obvious increase trend, showing better growth effect. After 12 days, the pH of samples under different $Na_2CO_3$ concentrations all decreased, which was due to the microalgae cells entering the lysis phase because of the depletion of nutrients. The photosynthesis weakened while respiration was enhanced, leading the produced $CO_2$ dissolving in water to cause the pH of the culture solution to decline, so the environment was not conducive to microalgae growth, and microalgae cells decay was aggravated. It could be observed that the colour of the microalgae liquid changed from green to yellow.

The DIC concentration changing with microalgae growth is shown in figure 4. It can be seen that the DIC concentration generally showed a trend of decreasing firstly and then increasing. The decrease of DIC concentration was the direct evidence that *S. obliquus* used IC in the solution to compose biomass. In detail, when the $Na_2CO_3$ concentration was 20 mg $l^{-1}$, the decrease of DIC could last for

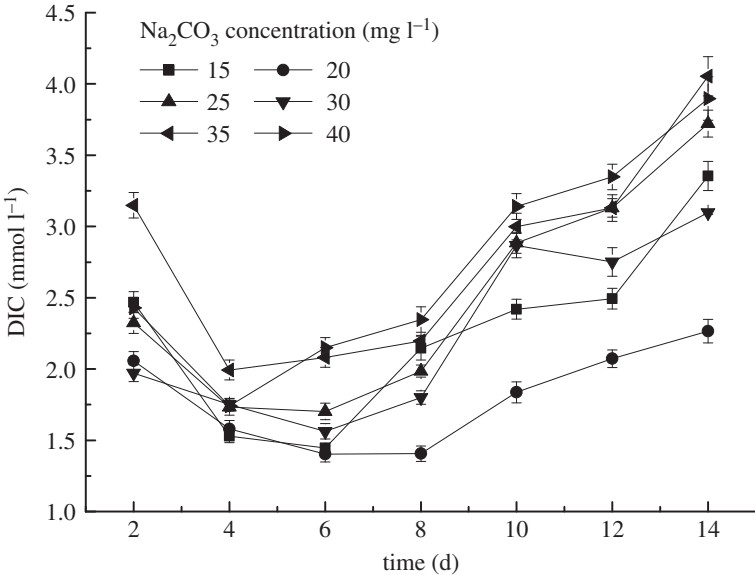

**Figure 4.** The DIC concentration changes of the microalgae liquid under the different $Na_2CO_3$ concentrations.

longer time. Besides, the fixed rate of IC at a $Na_2CO_3$ concentration of 15 mg l$^{-1}$ was 41.5%. By contrast, the fixation rate of IC was lower at higher $Na_2CO_3$ concentration, ranging from 20 to 30%. Therefore, according to the observed rules of IC recycle, supplying different concentration of IC to microalgae culture system in practical application could achieve different goals. For example, in order to promote the growth of microalgae to obtain more biomass, the best concentration suitable for the growth of microalgae could be adopted. If the purpose was achieving a higher utilization rate of IC to remove pollutants from wastewater and exhaust gas, pollutants should be supplied to the microalgae culture system at a lower concentration. Correspondingly, high concentration pollutants could be treated using microalgae solution of high concentration.

The reason why DIC concentrations decreased firstly and then increased was that the light and dark time was controlled in the short term. The photosynthesis intensity varies greatly, while the $CO_2$ contained in the water is relatively not high and diffused slowly at the gas–water interface and in the water [22]. The $CO_2$ supplementation in the water from atmosphere was limited, so photosynthesis played a dominant role in the change of DIC concentration in water in the short term. However, $CO_2$ in air continually dissolved, and sampling time was same of each day; accordingly, the change of photosynthesis carbon sequestration rate had little effect in the long run. In addition, at the later stage of the experiment, microalgae growth was limited because of depletion of nutrients in the water, the increase of microalgae cells density and the change in culture environment pH. However, the $CO_2$ from air was continuously replenished; therefore, the content of DIC had increased relative to the previous period.

## 3.3. Recycle of ammonia and phosphorus at different $Na_2CO_3$ concentrations

The changes of $NH_3$-N and $PO_4^{3-}$-P removal with time at different $Na_2CO_3$ concentrations under autotrophic conditions are exhibited in figure 5. Corresponding with the growth curve, $NH_3$-N and $PO_4^{3-}$-P contaminants were dramatically decreased within 4 days throughout the experiments. Later on, the removal in the treatment slowed down at the end of the experiment. The ammonia and phosphorus concentration dramatically decreased firstly, because nutrient richness in the medium and microalgae was in exponential phase at the beginning, and then started to level off due to exhaustion of nutrients. Nitrogen is an important nutrient for microalgae cells and a basic component of phytoplankton cell proteins, enzymes, nucleic acids and chlorophyll. Ammonia concentration decreased from 151 mg l$^{-1}$ to 85, 72, 78, 83, 79 and 74 mg l$^{-1}$, respectively. Regarding the effect of additional $Na_2CO_3$ on the pH of the solution, ammonia elimination was greatly influenced by pH of the culture medium, according to the following chemical equation (3.3):

$$NH_4^+ + OH^- \Leftrightarrow NH_3 + H_2O. \tag{3.3}$$

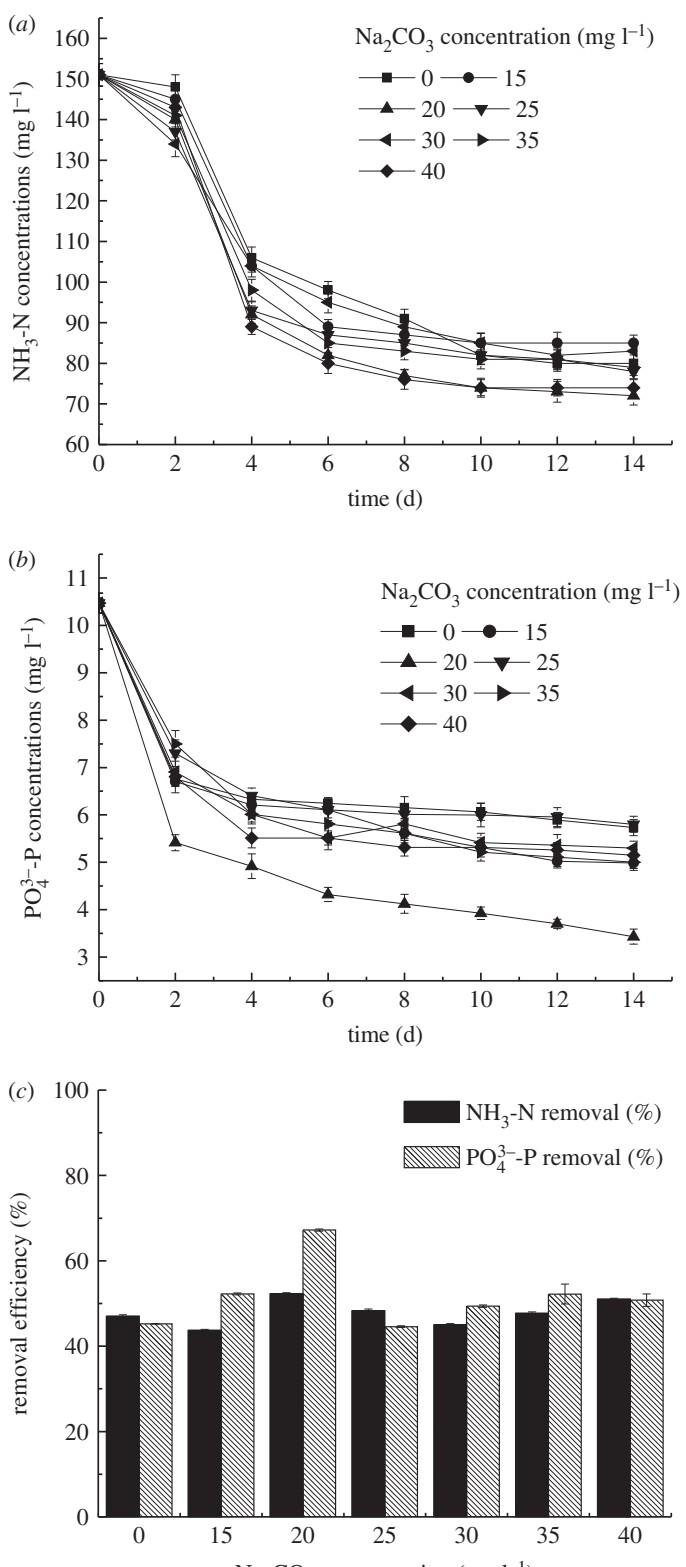

**Figure 5.** Nutrients removal profiles of *S. obliquus* at different $Na_2CO_3$ concentrations under autotrophic conditions (*a*) time course profiles of $NH_3$-N concentration, (*b*) time course profiles of $PO_4^{3-}$-P concentration and (*c*) removal efficiency of $NH_3$-N and $PO_4^{3-}$-P.

Since pH was higher than 7.0 in the reaction, the equilibrium shifted to the $NH_3$ production, resulting in a slight decrease in the ammonia concentration in the wastewater [6]. $NH_4^+$ was used by reducing to $NH_3$. So it is necessary to pay attention to the importance of $NH_3$ desorption in the ammonium reduction from wastewater.

As seen in figure 5c, when the test came to an end, 44, 52, 48, 45, 48 and 51% of NH$_3$-N were accordingly recycled from the 15, 20, 25, 30, 35 and 40 mg l$^{-1}$ Na$_2$CO$_3$ cultures. It can be seen that the recycle of ammonia nitrogen from the different Na$_2$CO$_3$ concentrations was approximately 50%. So the Na$_2$CO$_3$ concentration was not in positive correlation with the recycle of ammonia nitrogen to some extent. The removal of ammonia nitrogen might also be related to the concentration of phosphorus.

According to Cao et al. [23], the main factors affecting nitrogen absorption are: nitrogen concentration, nitrogen type and nitrogen to phosphorus ratio (N : P). In this study, the amount of nitrogen was sufficient. The results showed that microalgae preferentially used ammonia nitrogen when treating wastewater [24,25]. NH$_4$Cl was used as nitrogen source in this experiment, so the inhibitory factor was N : P. Due to the relatively fixed N : P of algae cell composition, nitrogen and phosphorus can only be fully used under optimal N : P [26]. Rhee [27] reported that there is no growth limitation by N and P simultaneously. Growth is limited by P or N on either side of the optimum N : P. When below the optimal cell N : P, growth was determined solely by N limitation and above optimal cell N : P, by P limitation [28]. Ammonia nitrogen removal rate decreased with increasing N : P [29]. Phosphorus is the main component of intracellular nucleic acids, proteins and phospholipids, participates in various metabolisms during the growth of microalgae and plays an important role in the growth phase. When algae species are in an environment with low concentration of P, photosynthesis is inhibited, resulting in less biomass and inhibition of protein synthesis, thereby inhibiting nitrogen uptake. Luo et al. [30] suggested that biomass of S. obliquus was proportional to P concentration, and phosphorus deficiency led to low biomass and protein content decreased [31]. In this study, as phosphorus was depleted, cell growth was inhibited, and N : P was increased, so absorption of nitrogen was inhibited by the concentration of phosphorus.

The tendency for PO$_4^{3-}$-P recycle in all treatments was similar to that for NH$_3$-N recycle. Phosphorus is also an important nutrient in the normal growth of microalgae for synthesis protein, chlorophyll, membrane and polyphosphate. Microalgae absorb phosphorus excessively and accumulate in cells. It was observed that the 15, 25, 30, 35 and 40 mg l$^{-1}$ Na$_2$CO$_3$ media witnessed PO$_4^{3-}$-P concentration decrease in 4 days and subsequently got to a stable stage. At the beginning, besides absorption, there was also adsorption, but no longer obvious after reaching equilibrium, and the removal rate was reduced subsequently. When the Na$_2$CO$_3$ concentration was 20 mg l$^{-1}$, the phosphorus concentration ranged from 10.47 to 3.43 mg l$^{-1}$, meeting the emission standards, and continuously decreased until end of experiment, meaning more phosphorus were recycled. The results explained the subsequent fall in PO$_4^{3-}$-P values, with the increase in microalgae concentration. In the first 2 days, the concentration of phosphorus dropped significantly and the fall was not attributable to consumption by microalgae but rather to adsorption both to the surface of the cells and to the surface of the bioreactor [6]. At the end of the experiment, the P eliminations decreased apparently as a consequence of lack of nutrition and probably the imbalance of N/P ratio of the culture medium.

As seen in figure 5c, the removal efficiency of PO$_4^{3-}$-P was higher than that of NH$_3$-N, and 67% PO$_4^{3-}$-P in wastewater was recycled when the Na$_2$CO$_3$ concentration was 20 mg l$^{-1}$. Possibly, microalgae could use orthophosphate directly through the photosynthesis to its own cell components. On the other hand, in alkaline conditions, orthophosphate could be removed from water in the form of chemical precipitation produced by co-precipitation of phosphate and calcium and magnesium ions. The N/P ratio used in the study changed with time and could well support the growth of S. obliquus.

Besides microalgae, many aquatic plants and microorganisms are also used for nutrient removal from wastewater. Aquatic plants absorb nutrients through roots. The removal of NH$_3$-N and TP by Lemna aequinoctialis at different initial concentrations of Zn$^{2+}$ was studied and removal rates without Zn$^{2+}$ of NH$_3$-N and TP were 84 and 98%, respectively. When exposed to high Zn$^{2+}$ concentrations, the cells showed peroxidation damage which significantly inhibited the NH$_3$-N and TP removal [32]. Hu et al. [33] investigated Lemna aequinoctialis treatment NH$_3$-N and TP in ADSW contaminated by oxytetracycline (OTC), and results showed that the nutrient removal could be inhibited significantly with the increase of OTC concentrations. Cheng et al. [34] cultured Spirodela punctala in synthetic ADSW, and the removal rate of NH$_3$-N and PO$_4$-P was higher than 99.2% during the 16 day culture. Compared with other aquatic plants, microalgae have higher removal rates of nitrogen and phosphorus and can be used for biodiesel. In recent years, the application of photosynthetic bacteria in wastewater treatment has also attracted the attention of researchers. Under the condition that the mixing ratio of the two bacteria is 1 : 1, the chemical oxygen demand removal rate reached 83.3% [35].

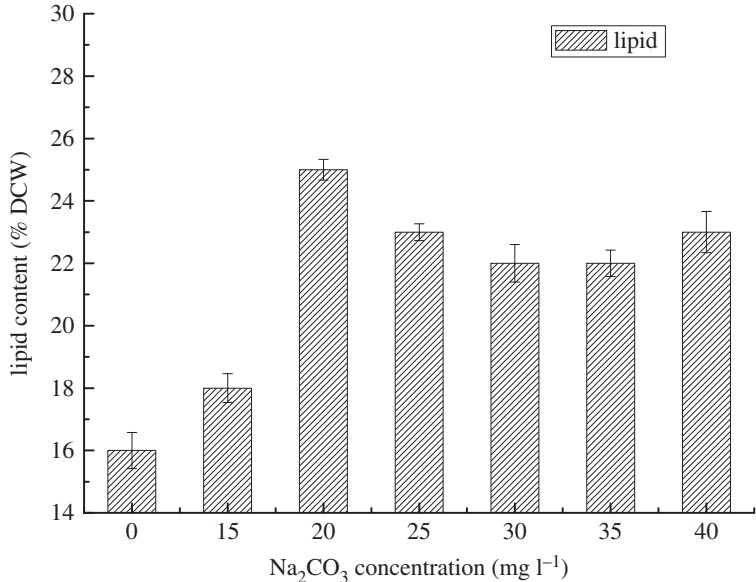

**Figure 6.** The lipid content of *S. obliquus* at six Na$_2$CO$_3$ concentrations under autotrophic condition (DCW, dry cell weight).

**Table 1.** Comparison of the fatty acid components of various microalgae with wastewater.

| microalgae strain | FAME type (% w/w) | | | | | references |
|---|---|---|---|---|---|---|
| | palmitate (C16:0) | palmitoleate (C16:1) | oleate (C18:1) | linoleate (C18:2) | linolenate (C18:3) | |
| *Coelastrella* sp. | 23.8–30.8 | — | — | 11.9–20.8 | 40.5–53.8 | [1] |
| *Scenedesmus abundans* | 17 | 7.5 | 37.5 | 10 | 6 | [38] |
| *Scenedesmus obliquus* | 26.1 | 3.3 | 49.2 | 5.7 | 5.5 | [39] |
| *Scenedesmus* sp. | 25.51 | 1.86 | 10.23 | 17.49 | — | [4] |

## 3.4. Lipid accumulation and microalgae harvest at different Na$_2$CO$_3$ concentrations

Microalgae varied in their proportions of protein (6–52%), carbohydrate (5–23%) and lipid (7–23%) [36]. The lipid content of *S. obliquus* in six different cultures were measured at the end of 14 days, as shown in figure 6. The results also suggested that there were slight differences between 25, 30, 35 and 40 mg l$^{-1}$ Na$_2$CO$_3$ concentrations ranging from 22 to 23%, while the highest lipid content was 25% at 20 mg l$^{-1}$ Na$_2$CO$_3$ concentration. The lowest was 18% at 15 mg l$^{-1}$, because the insufficient carbon source led to poor growth of microalgae. It was sufficient to add additional Na$_2$CO$_3$ concentration of 20 mg l$^{-1}$ on the basis of BG11 medium with 0.4 g l$^{-1}$ nitrogen and 0.077 g l$^{-1}$ phosphorus.

## 3.5. Potential value-added products of microalgae-based treatment

Fatty acid methyl esters (FAME), a lipid fraction of the algal biomass, could be used for biodiesel production and its main composition (C16–18) nearly reaching up to 92% was used as potential indicator of the biodiesel productivity [37]. Comparisons of the fatty acid components of various microalgae with wastewater are displayed in table 1. In terms of biodiesel quality, it is significant to promote the percentage of C16–18 ranging from microalgae species depending on different cultivation conditions for green biorefining. As for pigment, a main photosynthetic pigment can be obtained on microalgae feedstock, such as *S. obliquus*. Moreover, lutein is consumed as a food additive to prevent or ameliorate cardiovascular diseases, some types of cancer and age-related diseases [40]. Eicosapentaenoic acid, an omega-3 poly unsaturated fatty acid, playing a pivotal role in human health and used in nutraceutical products, can be produced by the algae *Nannochloropsis oceanica* with

sufficient carbon dioxide supply [41]. In addition to biofuel, bioenergy (ethanol and hydrogen) production from algal biomass, prior to food crop, forest residue and waste, can be enhanced after long-term ultrasonication pretreatment in an anaerobic environment [42]. More value-added products can be studied by taking advantage of lipid accumulation and it will be the trend of development to integrate algae-based process for wastewater treatment.

## 4. Conclusion

This study investigated the nutrient recycle and the multiplication of *S. obliquus* in synthetic wastewater with different concentrations of $Na_2CO_3$ added, ranging from 15 to 40 mg l$^{-1}$.

(1) The changes of pH and DIC were direct evidence that *S. obliquus* could use $Na_2CO_3$ to grow, and IC fixation rate was inversely proportional to its concentration. The specific growth rates of microalgae reached the maximum at day 2, corresponding with the rapid recycle rate of $NH_3$-N and $PO_4^{3-}$-P.

(2) The optimum concentration was 20 mg l$^{-1}$, with the highest lipid content of 25% and the biomass of 0.21 g l$^{-1}$. Moreover, the recycle rate of $NH_3$-N and $PO_4^{3-}$-P in the condition of 20 mg l$^{-1}$ $Na_2CO_3$ was highest during the process. Thus, the IC source added appropriately can improve the effectiveness.

(3) Besides, biomass could be in positive correlation with the $Na_2CO_3$ concentration except 20 mg l$^{-1}$, making a possibility that the *S. obliquus* could be acclimatized to adjust the condition of high IC concentration and produce more biomass.

Therefore, the growth of *S. obliquus* in synthetic wastewater offers great promise for the treatment of wastewater and production of renewable bioenergy.

Data accessibility. This article does not contain any additional data.
Authors' contributions. J.Y. conceived and designed the work. M.Z. and Y.L. assisted in collecting and compiling the resource materials and in manuscript preparation. X.G. conducted the research work, and then wrote the manuscript. Y.D. revised the manuscript. All authors read and approved the final manuscript to be published.
Competing interests. There are no competing interests to declare
Acknowledgements. The authors gratefully acknowledge Taiyuan University of Technology for providing experimental conditions. We are also thankful to C. Chen, who gave important advice for the revised article.
Funding. The research was funded by the Key Research and Development Project in Shanxi Province (201603D221028-1), which has facilitated the work.

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
