## [Reviewer comments · Royal Society Open Science]

Review History

RSOS-191214.R0 (Original submission)

Review form: Reviewer 1

Is the manuscript scientifically sound in its present form?

No

Are the interpretations and conclusions justified by the results?

Yes

Is the language acceptable?

Yes

Do you have any ethical concerns with this paper?

No

Have you any concerns about statistical analyses in this paper?

Yes

Recommendation?

Reject

Comments to the Author(s)

This paper is concerned to determine a potential contribution of *Scenedesmus obliquus* and recycle of nutrients in wastewater combined with inorganic carbon under autotrophic conditions. However, the manuscript is purely descriptive analytical protocols does not report on significant advances in scientific research. I have considered it does not bring about sufficient information to justify publication. For all these reasons I do not recommend its publication in Environmental Technology in the present state.

Review form: Reviewer 2**Is the manuscript scientifically sound in its present form?**

No

Are the interpretations and conclusions justified by the results?

Yes

Is the language acceptable?

Yes

Do you have any ethical concerns with this paper?

No

Have you any concerns about statistical analyses in this paper?

No

Recommendation?

Major revision is needed (please make suggestions in comments)

Comments to the Author(s)

General Comments

This manuscript investigates the role of different concentrations of Na_2CO_3 as inorganic carbon (IC) source for the growth of *Scenedesmus obliquus*. The paper is potentially interesting for the readers of the Journal and could be considered for publication after changes.

Specific Comments

1. Title of the paper. Since the substrate used was BG-11 supplemented with nutrients the term "synthetic wastewater" should be used instead of wastewater.
2. Section 3.1. Please check the composition of BG-11 medium. It seems that NaNO_3 and ferric ammonium citrate are different from the typical BG-11 composition. If this the case, the BG-11 medium should be mentioned as modified BG-11.
3. Section 3.1. Describe the illumination system.
4. Section 3.2. Please describe how many replicates of every sodium carbonate concentration were conducted. A standard deviation should be provided in the figures.
5. Section 3.2. Please clarify that the final concentration of sodium carbonate was between 15 and 40 mg/L.
6. Section 3.3. The authors mentioned that a 10-mL sample volume was withdrawn each sampling day. Based on the above, it seems that at the end of the experiment less than the half-volume of

- culture was left in the flask. What is the effect of the above procedure on the culture ? Furthermore, did authors experience problems with water evaporation in the flasks ? Please discuss it.
7. Section 4.1. For the cultivation experiments an initial biomass concentration of around 60 mg/L was used. What is the corresponding cell number ?
8. Section 4.1. No lag phase was observed in all sodium carbonate concentrations used. Please explain it.
9. Page 5, line 1. Please provide a reference for the optimal C/N ratio.
10. Figure 4. The initial ammonia and phosphorus concentration does not correspond to the quantity added to the medium.
11. Section 4.4. The second paragraph should be removed, since the harvest process was not the aim of the study.
12. Section 4.5. What was the FAME's composition in the present study ?

Review form: Reviewer 3

Is the manuscript scientifically sound in its present form?

Yes

Are the interpretations and conclusions justified by the results?

Yes

Is the language acceptable?

Yes

Do you have any ethical concerns with this paper?

No

Have you any concerns about statistical analyses in this paper?

No

Recommendation?

Major revision is needed (please make suggestions in comments)

Comments to the Author(s)

In this manuscript, the authors investigated the nutrients recycle and the growth of *Scenedesmus obliquus* in wastewater under different sodium carbonate concentrations. The topic is interesting, and overall, the methods and the results were presented satisfactorily. However, some parts of the manuscript were not presented well, and revisions are needed prior to a possible publication in Royal Society Open Science. Detailed comments are listed as follows.

On Introduction

1. In the Introduction, the current state in the research field should be presented with more details, and the authors should describe more recent progresses on the response of microalgae and plants to pollutants, for example, data and results described in the following articles: Responses of microalgae *Coelastrrella* sp. to stress of cupric ions in treatment of anaerobically digested swine wastewater, *Bioresource Technology*, 2018, 251: 274-279; Effects of copper ions on removal of nutrients from swine wastewater and release of dissolved organic matters in duckweed systems, *Water Research*, 2019, 158: 171-181.

On Material and Methods

2.3.2. Preparation of synthetic wastewater: The authors should set up a control group, and indicate the initial concentration of NH₃-N and PO₄ – P in synthetic wastewater.

3. The author could examine and discuss more on the content of enzymes or other physiological and biochemical properties to reflect the effects of Na₂CO₃ on microalgae growth either in this section or later.

On Results and Discussion

4. Error bars should be provided in the figures whenever available, otherwise, the authors should explain the reasons and effect on data credibility.

5.4.2 Absorption and utilization of Na₂CO₃ by photosynthesis of microalgae: Why the carbon sequestration of photosynthesis could increase the pH?

6.4.3. Recycle of ammonia and phosphorus at different Na₂CO₃ concentrations: The concentration of ammonia in the solution was still at 80 mg L⁻¹ after 6 days, why the authors said that the removal efficiency of ammonia was restricted by the exhaustion of nutrients? At the end of this work, whether the ammonia concentrations are meeting the emission standards?

7.4.3. Recycle of ammonia and phosphorus at different Na₂CO₃ concentrations: Besides microalgae, many aquatic plants and microorganisms are also used for the nutrient removal from the high nitrogen and phosphorus wastewater, such as duckweed and bacteria. What is the difference between microalgae and their removal efficiency? The following literature and the article mentioned above could serve this purpose on some aspects: Phytoremediation of anaerobically digested swine wastewater contaminated by oxytetracycline via *Lemna aequinoctialis*: Nutrient removal, growth characteristics and degradation pathways, *Bioresource Technology*, 2019, 291, 121853; Effect of zinc ions on nutrient removal and growth of *Lemna aequinoctialis* from anaerobically digested swine wastewater, *Bioresource Technology*, 2018, 249: 457-463; Treatment of anaerobically digested swine wastewater by *Rhodobacter blasticus* and *Rhodobacter capsulatus*, *Bioresource Technology*, 2016, 222, 33-38.

8.4.3. Recycle of ammonia and phosphorus at different Na₂CO₃ concentrations: The authors should explain why the removal of ammonia nitrogen is related to the concentration of phosphorus in the wastewater?

9.4.3. Recycle of ammonia and phosphorus at different Na₂CO₃ concentrations: What means about "At the beginning, besides absorption, there was also adsorption."?

10.4.4. Lipid accumulation and microalgae harvest at different Na₂CO₃ concentrations: The second paragraph of this section has nothing to do with the results of this study, please delete it.

11.4.5. Potential value added products of microalgae-based treatment: The authors should investigate the fatty acid methyl esters (FAME) composition under different Na₂CO₃ concentrations.

On Conclusion

12. In Conclusion, the authors should provide a concise illustration of the important results of this work. Please re-edit the Conclusion.

13. The authors should indicate the deficiency of this work and propose the possible solution methods.
14. The authors should edit and revise this submission more carefully and thoroughly.

Decision letter (RSOS-191214.R0)

02-Sep-2019

Dear Dr Duan:

Title: Nutrients recycle and the growth of *Scenedesmus obliquus* in wastewater under different sodium carbonate concentrations
Manuscript ID: RSOS-191214

The editor assigned to your manuscript has now received comments from reviewers. We would like you to revise your paper in accordance with the referee and Subject Editor suggestions which can be found below (not including confidential reports to the Editor). Please note this decision does not guarantee eventual acceptance.

Please submit your revised paper before 25-Sep-2019. Please note that the revision deadline will expire at 00.00am on this date. If we do not hear from you within this time then it will be assumed that the paper has been withdrawn. In exceptional circumstances, extensions may be possible if agreed with the Editorial Office in advance. We do not allow multiple rounds of revision so we urge you to make every effort to fully address all of the comments at this stage. If deemed necessary by the Editors, your manuscript will be sent back to one or more of the original reviewers for assessment. If the original reviewers are not available we may invite new reviewers.

RSC Associate Editor:
Comments to the Author:
(There are no comments.)

RSC Subject Editor:
Comments to the Author:
(There are no comments.)

Reviewers' Comments to Author:
Reviewer: 1

Comments to the Author(s)

This paper is concerned to determine a potential contribution of *Scenedesmus obliquus* and recycle of nutrients in wastewater combined with inorganic carbon under autotrophic conditions. However, the manuscript is purely descriptive analytical protocols does not report on significant advances in scientific research. I have considered it does not bring about sufficient information to justify publication. For all these reasons I do not recommend its publication in *Environmental Technology* in the present state.

Reviewer: 2

Comments to the Author(s)

General Comments

This manuscript investigates the role of different concentrations of Na_2CO_3 as inorganic carbon (IC) source for the growth of *Scenedesmus obliquus*. The paper is potentially interesting for the readers of the *Journal* and could be considered for publication after changes.

Specific Comments

1. Title of the paper. Since the substrate used was BG-11 supplemented with nutrients the term "synthetic wastewater" should be used instead of wastewater.
2. Section 3.1. Please check the composition of BG-11 medium. It seems that NaNO_3 and ferric ammonium citrate are different from the typical BG-11 composition. If this the case, the BG-11 medium should be mentioned as modified BG-11.
3. Section 3.1. Describe the illumination system.
4. Section 3.2. Please describe how many replicates of every sodium carbonate concentration were conducted. A standard deviation should be provided in the figures.

5. Section 3.2. Please clarify that the final concentration of sodium carbonate was between 15 and 40 mg/L.
6. Section 3.3. The authors mentioned that a 10-mL sample volume was withdrawn each sampling day. Based on the above, it seems that at the end of the experiment less than the half-volume of culture was left in the flask. What is the effect of the above procedure on the culture ? Furthermore, did authors experience problems with water evaporation in the flasks ? Please discuss it.
7. Section 4.1. For the cultivation experiments an initial biomass concentration of around 60 mg/L was used. What is the corresponding cell number ?
8. Section 4.1. No lag phase was observed in all sodium carbonate concentrations used. Please explain it.
9. Page 5, line 1. Please provide a reference for the optimal C/N ratio.
10. Figure 4. The initial ammonia and phosphorus concentration does not correspond to the quantity added to the medium.
11. Section 4.4. The second paragraph should be removed, since the harvest process was not the aim of the study.
12. Section 4.5. What was the FAME's composition in the present study ?

Reviewer: 3

Comments to the Author(s)

In this manuscript, the authors investigated the nutrients recycle and the growth of *Scenedesmus obliquus* in wastewater under different sodium carbonate concentrations. The topic is interesting, and overall, the methods and the results were presented satisfactorily. However, some parts of the manuscript were not presented well, and revisions are needed prior to a possible publication in Royal Society Open Science. Detailed comments are listed as follows.

On Introduction

1. In the Introduction, the current state in the research field should be presented with more details, and the authors should describe more recent progresses on the response of microalgae and plants to pollutants, for example, data and results described in the following articles: Responses of microalgae *Coelastrella* sp. to stress of cupric ions in treatment of anaerobically digested swine wastewater, *Bioresource Technology*, 2018, 251: 274-279; Effects of copper ions on removal of nutrients from swine wastewater and release of dissolved organic matters in duckweed systems, *Water Research*, 2019, 158: 171-181.

On Material and Methods

2.3.2. Preparation of synthetic wastewater: The authors should set up a control group, and indicate the initial concentration of $\text{NH}_3\text{-N}$ and $\text{PO}_4\text{-P}$ in synthetic wastewater.

3. The author could examine and discuss more on the content of enzymes or other physiological and biochemical properties to reflect the effects of Na_2CO_3 on microalgae growth either in this section or later.

On Results and Discussion

4. Error bars should be provided in the figures whenever available, otherwise, the authors should explain the reasons and effect on data credibility.

5.4.2 Absorption and utilization of Na_2CO_3 by photosynthesis of microalgae: Why the carbon sequestration of photosynthesis could increase the pH?

6.4.3. Recycle of ammonia and phosphorus at different Na_2CO_3 concentrations: The concentration of ammonia in the solution was still at 80 mg L⁻¹ after 6 days, why the authors said that the removal efficiency of ammonia was restricted by the exhaustion of nutrients? At the end of this work, whether the ammonia concentrations are meeting the emission standards?

7.4.3. Recycle of ammonia and phosphorus at different Na_2CO_3 concentrations: Besides microalgae, many aquatic plants and microorganisms are also used for the nutrient removal from the high nitrogen and phosphorus wastewater, such as duckweed and bacteria. What is the difference between microalgae and their removal efficiency? The following literature and the article mentioned above could serve this purpose on some aspects: Phytoremediation of anaerobically digested swine wastewater contaminated by oxytetracycline via *Lemna aquinoctialis*: Nutrient removal, growth characteristics and degradation pathways, *Bioresource Technology*, 2019, 291, 121853; Effect of zinc ions on nutrient removal and growth of *Lemna aquinoctialis* from anaerobically digested swine wastewater, *Bioresource Technology*, 2018, 249: 457-463; Treatment of anaerobically digested swine wastewater by *Rhodobacter blasticus* and *Rhodobacter capsulatus*, *Bioresource Technology*, 2016, 222, 33-38.

8.4.3. Recycle of ammonia and phosphorus at different Na_2CO_3 concentrations: The authors should explain why the removal of ammonia nitrogen is related to the concentration of phosphorus in the wastewater?

9.4.3. Recycle of ammonia and phosphorus at different Na_2CO_3 concentrations: What means about "At the beginning, besides absorption, there was also adsorption."?

10.4.4. Lipid accumulation and microalgae harvest at different Na_2CO_3 concentrations: The second paragraph of this section has nothing to do with the results of this study, please delete it.

11.4.5. Potential value added products of microalgae-based treatment: The authors should investigate the fatty acid methyl esters (FAME) composition under different Na_2CO_3 concentrations.

On Conclusion

12. In Conclusion, the authors should provide a concise illustration of the important results of this work. Please re-edit the Conclusion.

13. The authors should indicate the deficiency of this work and propose the possible solution methods.

14. The authors should edit and revise this submission more carefully and thoroughly.

Author's Response to Decision Letter for (RSOS-191214.R0)

See Appendix A.

Decision letter (RSOS-191214.R1)

05-Nov-2019

Dear Dr Duan:

Title: Nutrients recycle and the growth of *Scenedesmus obliquus* in wastewater under different sodium carbonate concentrations

Manuscript ID: RSOS-191214.R1

The editor assigned to your paper has now received comments from reviewers. We would like you to revise your paper in accordance with the referee and Subject Editor suggestions which can be found below (not including confidential reports to the Editor). Please note this decision does not guarantee eventual acceptance.

Please submit a copy of your revised paper before 28-Nov-2019. Please note that the revision deadline will expire at 00.00am on this date. If we do not hear from you within this time then it will be assumed that the paper has been withdrawn. In exceptional circumstances, extensions may be possible if agreed with the Editorial Office in advance. We do not allow multiple rounds of revision so we urge you to make every effort to fully address all of the comments at this stage. If deemed necessary by the Editors, your manuscript will be sent back to one or more of the original reviewers for assessment. If the original reviewers are not available we may invite new reviewers.

RSC Associate Editor
Comments to the Author:
Please respond to the comments of reviewer 1.

Reviewers' Comments to Author:

Author's Response to Decision Letter for (RSOS-191214.R1)

See Appendix B.

Decision letter (RSOS-191214.R2)

19-Nov-2019

Dear Dr Duan:

Title: Nutrients recycle and the growth of *Scenedesmus obliquus* in wastewater under different sodium carbonate concentrations
Manuscript ID: RSOS-191214.R2

It is a pleasure to accept your manuscript in its current form for publication in Royal Society Open Science. The chemistry content of Royal Society Open Science is published in collaboration with the Royal Society of Chemistry.

RSC Associate Editor
Comments to the Author:
(There are no comments.)

Reviewer(s)' Comments to Author:

Appendix A

Dear Editors and reviewers:

Thank you for your letter dated 02 September. We were pleased to know that our work was rated as potentially acceptable for publication in journal of Royal Society Open Science. We appreciate the reviewers for the time and effort that they have put into reviewing the previous version of the manuscript. Their suggestions have enabled us to improve our work. Based on the instructions provided in your letter, we would like to resubmit the manuscript (RSOS-191214) entitled “Nutrients recycle and the growth of *Scenedesmus obliquus* in wastewater under different sodium carbonate concentrations”, with all the changes highlighted by using different color (red). There are no conflicts of interest in manuscript and all authors are in agreement with the content of the manuscript. This work is not under active consideration for publication elsewhere, has not been accepted for publication, nor has it been published.

Point-by-point responses to the reviewers’ comments are enclosed after the letter for your consideration.

We deeply appreciate your consideration of our resubmitted manuscript and we look forward to receiving comments from the reviewers.

Thank you and best regards!

Sincerely

Yun Duan

Corresponding author:

E-mail address: duanyun@tyut.edu.cn.

Tel. /fax: +863543176586

Address: College of Environmental Science and Engineering, Taiyuan University of
Technology, Taiyuan, Shanxi, 030024, China

Reviewers' comments to the Author:

Reviewer: 1

Comments to the Author(s)

This paper is concerned to determine a potential contribution of *Scenedesmus obliquus* and recycle of nutrients in wastewater combined with inorganic carbon under autotrophic conditions. However, the manuscript is purely descriptive analytical protocols does not report on significant advances in scientific research. I have considered it does not bring about sufficient information to justify publication. For all these reasons I do not recommend its publication in Environmental Technology in the present state.

Reviewer: 2

Comments to the Author(s)

General Comments

This manuscript investigates the role of different concentrations of Na_2CO_3 as inorganic carbon (IC) source for the growth of *Scenedesmus obliquus*. The paper is potentially interesting for the readers of the Journal and could be considered for publication after changes.

Specific Comments

1. Title of the paper. Since the substrate used was BG-11 supplemented with nutrients the term "synthetic wastewater" should be used instead of wastewater.

Author's responses:

We modified the title to "Nutrients recycle and the growth of *Scenedesmus obliquus* in synthetic wastewater under different sodium carbonate concentrations" according to the comment (Page 1).

2. Section 3.1. Please check the composition of BG-11 medium. It seems that NaNO₃ and ferric ammonium citrate are different from the typical BG-11 composition. If this the case, the BG-11 medium should be mentioned as modified BG-11.

Author's responses:

According to your comment, we checked the composition of BG11 carefully. The concentration of NaNO₃ was corrected to 1500 mg L⁻¹ and the concentration of ferric ammonium citrate was confirmed there are no errors.(Section 3.1, Line 2)

3. Section 3.1. Describe the illumination system.

Author's responses:

The illumination system was described in detail.(Section 3.1, Line 9)

4. Section 3.2. Please describe how many replicates of every sodium carbonate concentration were conducted. A standard deviation should be provided in the figures.

Author's responses:

Experiment were set up three parallel experiment for each concentration. A standard deviation was provided in the figures in this revision .

5. Section 3.2. Please clarify that the final concentration of sodium carbonate was between 15 and 40 mg/L.

Author's responses:

The final concentration of sodium carbonate was clarified in the text.

6. Section 3.3. The authors mentioned that a 10-mL sample volume was withdrawn each sampling day. Based on the above, it seems that at the end of the experiment less than the half-volume of culture was left in the flask. What is the effect of the above procedure on the culture ? Furthermore, did authors experience problems with water evaporation in the flasks ? Please discuss it.

Author's responses:

We shake the algae solution before sampling to keep the ratio of microalgae to nutrients constant, and minimize the impact of sampling on later culture. We have learned about the knowledge of evaporation. Considering that the mouth of the flask is covered with a sealing film and cultured in the incubator, the temperature is suitable and constant and the air flow is weak, and the influence of evaporation is considered to be minimal.

7. Section 4.1. For the cultivation experiments an initial biomass concentration of around 60 mg/L was used. What is the corresponding cell number ?

Author's responses:

The initial biomass concentration was 33×10^5 cells mL⁻¹ and has been added in the text.(Section 3.1, Line 7)

8. Section 4.1. No lag phase was observed in all sodium carbonate concentrations used. Please explain it.

Author's responses: We think the main reasons why there was no lag phase are as follows: Firstly, the microalgae used in the experiment were pre-cultured to logarithmic growth period, and the growth ability is the most exuberant. Secondly, The carbon source phosphorus source is the same as the original BG11 medium, and the nitrogen source is the ammonia nitrogen which is more suitable type for the growth of microalgae. The other experimental conditions were completely consistent with the pre-culture conditions. It's also possible that there may have been a brief lag phase, and our measurements began two days after inoculation, so we didn't observe it.

9. Page 5, line 1. Please provide a reference for the optimal C/N ratio.

Author's responses:

Reference for the optimal C/N ratio was marked in the text.

10. Figure 4. The initial ammonia and phosphorus concentration does not correspond to the quantity added to the medium.

Author's responses:

The initial nitrogen and phosphorus concentration was a measured value and higher than the added amount. It is assumed that the main cause of this phenomenon is that the inoculated microalgae solution was cultured for a period of time, and the extracellular polymeric substances (EPS) produced adheres to the cell surface. The main components of EPS are nucleic acids, proteins,etc.,containing nitrogen and phosphorus, the ferric citrate also contains ammonium. In addition, ammonium ferric citrate also contains ammonium, which together cause the ammonia nitrogen concentration to be greater than the ammonia nitrogen addition. The above factors together lead to initial ammonia and phosphorus concentration does not correspond to the quantity added to the medium.

11. Section 4.4. The second paragraph should be removed, since the harvest process was not the aim of the study.

Author's responses:

The second paragraph of section 4.4 was removed.

12. Section 4.5. What was the FAME's composition in the present study ?

Author's responses:

We didn't investigate the FAME's composition at present, which could be presented in later article.

Reviewer: 3

Comments to the Author(s)

In this manuscript, the authors investigated the nutrients recycle and the growth of *Scenedesmus obliquus* in wastewater under different sodium carbonate concentrations. The topic is interesting, and overall, the methods and the results were presented satisfactorily. However, some parts of the manuscript were not presented well, and revisions are needed prior to a possible publication in Royal Society Open Science. Detailed comments are listed as follows.

On Introduction

1. In the Introduction, the current state in the research field should be presented with more details, and the authors should describe more recent progresses on the response of microalgae and plants to pollutants, for example, data and results described in the following articles: Responses of microalgae *Coelastrrella sp.* to stress of cupric ions in treatment of anaerobically digested swine wastewater, *Bioresource Technology*, 2018, 251: 274-279; Effects of copper ions on removal of nutrients from swine wastewater and release of dissolved organic matters in duckweed systems, *Water Research*, 2019, 158: 171-181.

Author's responses:

Thank you for your recommendation. We have carefully read the above articles and supplemented this article.

On Material and Methods

2.3.2. Preparation of synthetic wastewater: The authors should set up a control group, and indicate the initial concentration of $\text{NH}_3\text{-N}$ and $\text{PO}_4^{3-}\text{-P}$ in synthetic wastewater.

Author's responses:

The initial concentration of $\text{NH}_3\text{-N}$ and $\text{PO}_4^{3-}\text{-P}$ in synthetic wastewater measured to be 151 mg L^{-1} , 10.46 mg L^{-1} , respectively, and it has been indicated in text.

3. The author could examine and discuss more on the content of enzymes or other physiological and biochemical properties to reflect the effects of Na_2CO_3 on microalgae growth either in this section or later.

Author's responses:

Considering your constructive advice, we are carrying out the examination of the content of enzymes, chlorophyll, protein and polysaccharide to reflect the effects of Na_2CO_3 on the microalgae growth, which could present in later article.

On Results and Discussion

4. Error bars should be provided in the figures whenever available, otherwise, the authors should explain the reasons and effect on data credibility.

Author's responses:

Error bars was provided in the figures in this revision.

5.4.2 Absorption and utilization of Na_2CO_3 by photosynthesis of microalgae: Why the carbon sequestration of photosynthesis could increase the pH?

Author's responses:

Because in weakly alkaline media, the IC forms that microalgae can utilize are mainly HCO_3^- and CO_2 dissolved in water. In particular, the increase of pH is owing to the absorption of HCO_3^- by microalgae, leading to OH^- accumulation. The mechanism is as follows:

where CA represents the extracellular carbonic anhydrase

Besides the consumption of IC, the absorption of other nutrients can also lead to the alkalization of the solution environment. The majority essential nutrients existed in ionic form, such as HCO_3^- , NO_3^- , and H_2PO_4^- , which assimilated H^+ from culture medium and released OH^- , thereby leading to pH increase.

The related content is detailed in the text. (Section 4.2, third paragraph)

6.4.3. Recycle of ammonia and phosphorus at different Na_2CO_3 concentrations: The concentration of ammonia in the solution was still at 80 mg L^{-1} after 6 days, why the authors said that the removal efficiency of ammonia was restricted by the exhaustion of nutrients? At the end of this work, whether the ammonia concentrations are meeting the emission standards?

Author's responses:

The removal efficiency of ammonia was mainly restricted by the exhaustion of phosphorus and the reasons have been fully explained in the text. (Last line 3, page 4). The ammonia concentrations are not meeting the emission standards.

7.4.3. Recycle of ammonia and phosphorus at different Na_2CO_3 concentrations: Besides microalgae, many aquatic plants and microorganisms are also used for the

nutrient removal from the high nitrogen and phosphorus wastewater, such as duckweed and bacteria. What is the difference between microalgae and their removal efficiency? The following literature and the article mentioned above could serve this purpose on some aspects: Phytoremediation of anaerobically digested swine wastewater contaminated by oxytetracycline via *Lemna aequinoctialis*: Nutrient removal, growth characteristics and degradation pathways, Bioresource Technology, 2019, 291, 121853; Effect of zinc ions on nutrient removal and growth of *Lemna aequinoctialis* from anaerobically digested swine wastewater, Bioresource Technology, 2018, 249: 457-463; Treatment of anaerobically digested swine wastewater by *Rhodobacter blasticus* and *Rhodobacter capsulatus*, Bioresource Technology, 2016, 222, 33-38.

Author's responses:

Thank you for your recommendation. We have carefully read the above articles and supplemented this article. (Section 4.3, last paragraph).

8.4.3. Recycle of ammonia and phosphorus at different Na_2CO_3 concentrations: The authors should explain why the removal of ammonia nitrogen is related to the concentration of phosphorus in the wastewater?

Author's responses:

The reason why the removal of ammonia nitrogen is related to the concentration of phosphorus in the wastewater has been explained in text.(Last line 3, page 4)

9.4.3. Recycle of ammonia and phosphorus at different Na_2CO_3 concentrations: What means about "At the beginning, besides absorption, there was also adsorption."?

Author's responses:

It means that absorption and assimilation are used for the synthesis of intracellular substances, and there is also adsorption on the cell surface. There were related instructions in the text, at (Line 19, Page 5).

10.4.4. Lipid accumulation and microalgae harvest at different Na_2CO_3 concentrations: The second paragraph of this section has nothing to do with the results of this study, please delete it.

Author's responses:

The second paragraph of section 4.4 was deleted.

11.4.5. Potential value added products of microalgae-based treatment: The authors should investigate the fatty acid methyl esters (FAME) composition under different Na_2CO_3 concentrations.

Author's responses:

Considering your valuable advice, we are proceeding the investigation of the FAME composition under different Na_2CO_3 concentrations, which could be presented in later article.

On Conclusion

12.In Conclusion, the authors should provide a concise illustration of the important results of this work. Please re-edit the Conclusion.

Author's responses:

The conclusion was re-edited (Page 6).

13.The authors should indicate the deficiency of this work and propose the possible solution methods.

Author's responses:

We did not examine and discuss more on the content of enzymes or other physiological and biochemical properties which reflect the effects of Na_2CO_3 on microalgae growth.

Other researchers in our team are conducting further research which could be published in future articles.

14.The authors should edit and revise this submission more carefully and thoroughly.

Author's responses:

The resubmitted was revised carefully and thoroughly according to your comments.

Appendix B

RSC Associate Editor

Comments to the Author:

Please respond to the comments of reviewer 1.

Reviewers' comments to the Author:

Reviewer:1

Comments to the Author(s)

This paper is concerned to determine a potential contribution of *Scenedesmus obliquus* and recycle of nutrients in wastewater combined with inorganic carbon under autotrophic conditions. However, the manuscript is purely descriptive analytical protocols does not report on significant advances in scientific research. I have considered it does not bring about sufficient information to justify publication. For all these reasons I do not recommend its publication in Environmental Technology in the present state.

Authors' response:

Thank you for your review. According to your valuable comments, we have carefully revised the manuscript. All the changes made in the text was in red color.

In this manuscript, we aimed to find the relationship of dealing with wastewater and culturing algae. After monitoring the growth of algae *Scenedesmus obliquus* in synthetic wastewater, we found that the biomass was positive correlation with the dissolved inorganic carbon concentration $20 \text{ mg}\cdot\text{L}^{-1}$.According the results, we drew the conclusion that the acclimation of algae could enhance the biomass adaptation in wastewater, which provided a possibility that *Scenedesmus obliquus* could be acclimatized to adjust to high concentrations of inorganic carbon to promote biomass accumulation and recycle nutrients of carbon, nitrogen and phosphorus during the wastewater treatment.

Even the journal of **Environmental Technology** has similarities with the journal of **Royal Society Open Science**, though the later journal covers the entire range of science and allows to publish all the high-quality work it receives without the usual restrictions on scope, length or impact. We ask you for reconsideration of the publishing of this manuscript in the journal of **Royal Society Open Science**.